# Quadratic Decomposable Submodular Function Minimization

**Pan Li**
UIUC
panli2@illinois.edu

**Niao He**
UIUC
niaohe@illinois.edu

**Olgica Milenkovic**
UIUC
milenkov@illinois.edu

## Abstract

We introduce a new convex optimization problem, termed *quadratic decomposable submodular function minimization*. The problem is closely related to decomposable submodular function minimization and arises in many learning on graphs and hypergraphs settings, such as graph-based semi-supervised learning and PageRank. We approach the problem via a new dual strategy and describe an objective that may be optimized via random coordinate descent (RCD) methods and projections onto cones. We also establish the linear convergence rate of the RCD algorithm and develop efficient projection algorithms with provable performance guarantees. Numerical experiments in semi-supervised learning on hypergraphs confirm the efficiency of the proposed algorithm and demonstrate the significant improvements in prediction accuracy with respect to state-of-the-art methods.[1]

## 1 Introduction

Given $[N] = \{1, 2, ..., N\}$, a submodular function $F : 2^{[N]} \to \mathbb{R}$ is a set function that for any $S_1, S_2 \subseteq [N]$ satisfies $F(S_1) + F(S_2) \geq F(S_1 \cup S_2) + F(S_1 \cap S_2)$. Submodular functions are ubiquitous in machine learning as they capture rich combinatorial properties of set functions and provide useful regularization functions for supervised and unsupervised learning [1]. Submodular functions also have continuous Lovász extensions [2], which establish solid connections between combinatorial and continuous optimization problems.

Due to their versatility, submodular functions and their Lovász extensions are frequently used in applications such as learning on directed/undirected graphs and hypergraphs [3, 4], image denoising via total variation regularization [5, 6] and MAP inference in high-order Markov random fields [7]. In many optimization settings involving submodular functions, one encounters the convex program

$$\min_x \sum_{i \in [N]} (x_i - a_i)^2 + \sum_{r \in [R]} [f_r(x)]^p,$$

where $a \in \mathbb{R}^N$, $p \in \{1, 2\}$, and where for all $r$ in some index set $[R]$, $f_r$ stands for the Lovász extension of a submodular function $F_r$ that describes a combinatorial structure over the set $[N]$. For example, in image denoising, each parameter $a_i$ may correspond to the observed value of a pixel $i$, while the functions $[f_r(x)]^p$ may be used to impose smoothness constraints on pixel neighborhoods. One of the main difficulties in solving this optimization problem comes from the nondifferentiability of the second term: a direct application of subgradient methods leads to convergence rates as slow as $1/\sqrt{k}$, where $k$ denotes the number of iterations [8].

In recent years, the above described optimization problem with $p = 1$ has received significant interest in the context of *decomposable submodular function minimization* (DSFM) [9]. The motivation for

studying this particular setup is two-fold: first, solving the convex optimization problem directly recovers the combinatorial solution to the submodular min-cut problem $\min_{S \subseteq [N]} F(S)$, where $F(S) = \sum_{r \in [R]} F_r(S) - 2 \sum_{i \in S} a_i$ [10]; second, minimizing a submodular function decomposed into a sum of simpler components $F_r$, $r \in [R]$, is much easier than minimizing an unrestricted submodular function $F$ over a large set $[N]$. There are several milestone results for the DSFM problem: Jegelka et al. [11] first tackled the problem by considering its dual and proposed a solver based on Douglas-Rachford splitting. Nishihara et al. [12] established the linear convergence rate of alternating projection methods for solving the dual problem. Ene et al. [13, 14] presented linear convergence rates of coordinate descent methods and subsequently tightened the results via submodular flows. Pan et al. [15] improved those methods by leveraging incidence relations of the arguments of submodular function components.

Here, we focus on the other important case when $p = 2$; we refer to the underlying optimization problem as *quadratic DSFM* (QDSFM). QDSFM appears naturally in a wide spectrum of applications, including learning on graphs and hypergraphs, and in particular, semi-supervised learning and PageRank. It has also been demonstrated both theoretically [16] and empirically [4, 17] that employing regularization with quadratic terms offers significantly improved predictive performance when compared to the case when $p = 1$. Despite the importance of the QDSFM problem, its theoretical and algorithmic developments have not reached the same level of maturity as those for the DSFM problem. To the best of our knowledge, only a few reported works [17, 18] have provided solutions for specific instances of QDSFMs with sublinear convergence guarantees.

This work takes a substantial step towards solving the QDSFM problem in its most general form by developing a family of algorithms with *linear convergence rate* and *small iteration cost*, including the randomized coordinate descent (RCD) and alternative projection (AP) algorithms. Our contributions are as follows. First, we derive a new dual formulation for the QDSFM problem since an analogue of the dual transformation for the DSFM problem is not applicable. Interestingly, the dual QDSFM problem requires one to find the best approximation of a hyperplane via a product cone as opposed to a product polytope, encountered in the dual DSFM problem. Second, we develop a linearly convergent RCD (and AP) algorithm for solving the dual QDSFM. Because of the special underlying conic structure, new analytic approaches are needed to prove the weak strong convexity of the dual QDSFM, which essentially guarantees linear convergence. Third, we develop generalized Frank-Wolfe and min-norm-point methods for efficiently computing the conic projection required in each step of RCD (and AP) and provide a $1/k$-rate convergence analysis. Finally, we evaluate our methods on semi-supervised learning over hypergraphs using synthetic and real datasets, and demonstrate superior performance both in convergence rate and prediction accuracy compared to existing methods. We postpone all the detailed proofs and supplementary discussion to the full version of this paper.

## 2  Notation and Problem Formulation

For a submodular function $F$ defined over the ground set $[N]$, the *Lovász extension* is a convex function $f : \mathbb{R}^N \to \mathbb{R}$, defined for all $x \in \mathbb{R}^N$ according to

$$f(x) = \sum_{k=1}^{N-1} F(\{i_1, ..., i_k\})(x_{i_k} - x_{i_{k+1}}) + F([N])x_{i_N}, \tag{1}$$

where $x_{i_1} \geq x_{i_2} \geq \cdots \geq x_{i_N}$. For a vector $x \in \mathbb{R}^N$ and a set $S \subseteq [N]$, let $x(S) = \sum_{i \in [S]} x_i$ where $x_i$ is the component of $x$ in $i$th dimention. Then, the *base polytope* of $F$, denoted by $B$, is defined as

$$B = \{y \in \mathbb{R}^N | y(S) \leq F(S), \ \forall S \subset [N], \ y([N]) = F([N])\}. \tag{2}$$

Using the base polytope, the Lovász extension can also be written as $f(x) = \max_{y \in B} \langle y, x \rangle$.

We say that an element $i \in [N]$ is incident to $F$ if there exists a $S \subset [N]$ such that $F(S) \neq F(S \cup \{i\})$. Furthermore, we use $(x)_+$ to denote the function $\max\{x, 0\}$. Given a positive diagonal matrix $W \in \mathbb{R}^{N \times N}$ and a vector $x \in \mathbb{R}^N$, we define the $W$-norm according to $\|x\|_W = \sqrt{\sum_{i=1}^N W_{ii} x_i^2}$, and simply use $\| \cdot \|$ when $W = I$, the identity matrix. For an index set $[R]$, we denote the $R$-product of $N$-dimensional Euclidean spaces by $\otimes_{r \in [R]} \mathbb{R}^N$. A vector $y \in \otimes_{r \in [R]} \mathbb{R}^N$ is written as $(y_1, y_2, ..., y_R)$, where $y_r \in \mathbb{R}^N$ for all $r \in [R]$. The $W$-norm induced on $\otimes_{r \in [R]} \mathbb{R}^N$ equals $\|y\|_{I(W)} = \sqrt{\sum_{r=1}^R \|y_r\|_W^2}$. We reserve the symbol $\rho$ for $\max_{y_r \in B_r, \forall r} \sqrt{\sum_{r \in [R]} \|y_r\|_1^2}$.

Next, we formally state the QDSFM problem. Consider a collection of submodular functions $\{F_r\}_{r \in [R]}$ defined over the ground set $[N]$, and denote their Lovász extensions and base polytopes by $\{f_r\}_{r \in [R]}$ and $\{B_r\}_{r \in [R]}$, respectively. We use $S_r \subseteq [N]$ to denote the set of variables incident to $F_r$ and make the further assumption that the functions $F_r$ are normalized and nonnegative, i.e., that $F_r(\emptyset) = 0$ and $F_r \geq 0$. These two mild constraints are satisfied by almost all submodular functions that arise in practical applications. We consider the following minimization problem:

$$\text{QDSFM:} \qquad \min_{x \in \mathbb{R}^N} \|x - a\|_W^2 + \sum_{r \in [R]} [f_r(x)]^2, \qquad (3)$$

where $a \in \mathbb{R}^N$ is a given vector and $W \in \mathbb{R}^{N \times N}$ is a *positive diagonal* matrix. As an immediate observation, the problem has a unique solution, denoted by $x^*$, due to the strong convexity of (3).

## 3 Applications

We start by reviewing some important machine learning problems that give rise to QDSFM.

***Semi-supervised Learning*** (SSL) is a learning paradigm that allows one to utilize the underlying structure or distribution of unlabeled samples, whenever the information provided by labeled samples does not suffice for learning an inductive predictor [19, 20]. A standard setting for a $K$-class transductive learner is as follows: given $N$ data points $\{z_i\}_{i \in [N]}$, and labels for the first $l$ ($\ll N$) samples $\{y_i | y_i \in [K]\}_{i \in [l]}$, the learner is asked to infer the labels for all the remaining data points $i \in [N]/[l]$. The widely-used SSL problem with least squared loss requires one to solve $K$ regularization problems: for each class $k \in [K]$, set the scores of data points within the class to

$$\hat{x}^{(k)} = \arg\min_{x^{(k)}} \beta \|x^{(k)} - a^{(k)}\|^2 + \Omega(x^{(k)}),$$

where $a^{(k)}$ represents the information provided by the known labels, i.e., $a_i^{(k)} = 1$ if $y_i = k$, and $0$ otherwise, $\beta$ denotes a hyperparameter and $\Omega$ stands for a smoothness regularizer. The labels of the data points are inferred according to $\hat{y}_i = \arg\max_k \{\hat{x}_i^{(k)}\}$. For typical graph and hypergraph learning problems, $\Omega$ is often chosen to be a Laplacian regularizer constructed using $\{z_i\}_{i \in [N]}$ (see Table 1). In Laplacian regularization, each edge/hyperedge corresponds to one functional component in the QDSFM problem. Note that the variables may also be normalized with respect to their degrees, in which case the normalized Laplacian is used instead. For example, in graph learning, one of the terms in $\Omega$ assumes the form $w_{ij}(x_i/\sqrt{d_i} - x_j/\sqrt{d_j})^2$, where $d_i$ and $d_j$ correspond to the degrees of the vertex variables $i$ and $j$, respectively. It can be shown using some simple algebra that the normalization term reduces to the matrix $W$ used in the definition of the QDSFM problem (3).

| One component in $\Omega(x)$ | Description of the combinatorial structure | The submodular function |
|---|---|---|
| $w_r(x_i - x_j)^2$, $S_r = \{i, j\}$ | Graphs: Nearest neighbors [4]/Gaussian similarity [21] | $F_r(S) = \sqrt{w_{ij}}$ if $|S \cap \{i, j\}| = 1$ |
| $w_r \max_{i, j \in S_r} (x_i - x_j)^2$ | Hypergraphs: Categorical features [17] | $F_r(S) = \sqrt{w_r}$ if $|S \cap S_r| \in [1, |S_r| - 1]$ |
| $w_r \max_{(i,j) \in H_r \times T_r} (x_i - x_j)_+^2$ | Directed hypergraphs: citation networks [18] | $F_r(S) = \sqrt{w_r}$ if $|S \cap H_r| \geq 1$, $|([N]/S) \cap T_r| \geq 1$ |
| General $[f_r(x)]^2$ | Submodular Hypergraphs: Mutual Information [22, 23] | A symmetric submodular function |

Table 1: Laplacian regularization in semi-supervised learning. In the third column, whenever the stated conditions are not satisfied, it is assumed that $F_r = 0$. For directed hypergraphs, $H_r$ and $T_r$ are subsets of $S_r$ termed the head and the tail set. When $H_r = T_r = S_r$, one recovers the setting for undirected hypergraphs.

***PageRank*** (PR) is a well-known method used for ranking Web pages [24]. Web pages are linked and they naturally give rise to a graph $G = (V, E)$, where, without loss of generality, one may assume that $V = [N]$. Let $A$ and $D$ be the adjacency matrix and diagonal degree matrix of $G$, respectively. PR essentially finds a fixed point $p \in \mathbb{R}^N$ via the iterative procedure $p^{(t+1)} = (1 - \alpha)s + \alpha AD^{-1}p^{(t)}$, where $s \in \mathbb{R}^N$ is a fixed vector and $\alpha \in (0, 1]$. It is easy to verify that $p$ is a solution of the problem

$$\min_p \frac{(1 - \alpha)}{\alpha} \|p - s\|_{D^{-1}}^2 + (D^{-1}p)^T(D - A)(D^{-1}p) = \|x - a\|_W^2 + \sum_{ij \in E} (x_i - x_j)^2, \qquad (4)$$

where $x = D^{-1}p$, $a = D^{-1}s$ and $W = \frac{(1-\alpha)}{\alpha}D$. Obviously, (4) may be viewed as a special instance of the QDSFM problem. Note that the PR iterations on graphs take the form $D^{-\frac{1}{2}}(p^{(t+1)} - p^{(t)}) =$

$(1-\alpha)D^{-\frac{1}{2}}(s-p^{(t)}) - \alpha L(D^{-\frac{1}{2}}p^{(t)})$, where $L = I - D^{-\frac{1}{2}}AD^{-\frac{1}{2}}$ is the normalized Laplacian of the graph. The PR problem for hypergraphs is significantly more involved, and may be formulated using diffusion processes (DP) based on a normalized hypergraph Laplacian operator $L$ [25]. The underlying PR procedure reads as $\frac{dx}{dt} = (1-\alpha)(a-x) - \alpha L(x)$, where $x(t) \in \mathbb{R}^N$ is the potential vector at time $t$. Tracking this DP precisely for every time point $t$ is a difficult task which requires solving a densest subset problem [25]. However, the stationary point of this problem, i.e., a point $x$ that satisfies $(1-\alpha)(a-x) - \alpha L(x) = 0$ may be easily found by solving the optimization problem

$$\min_x (1-\alpha)\|x-a\|^2 + \alpha\langle x, L(x)\rangle.$$

The term $\langle x, L(x)\rangle$ matches the normalized regularization term for hypergraphs listed in Table 1, i.e., $\sum_r \max_{i,j \in S_r}(x_i/\sqrt{d_i} - x_j/\sqrt{d_j})^2$. Clearly, once again this leads to the QDSFM problem. The PR equation for directed or submodular hypergraphs can be stated similarly using the Laplacian operators described in [26, 23, 27]. The PR algorithm defined in this manner has many advantages over the multilinear PR method based on higher-order Markov chains [28], since it allows for arbitrarily large orders and is guaranteed to converge for any $\alpha \in (0, 1]$. In the full version of this paper, we will provide a more detailed analysis of the above described PR method.

# 4    Algorithms for Solving the QDSFM Problem

We describe next the *first known linearly convergent* algorithms for solving the QDSFM problem. To start with, observe that the QDSFM problem is convex since the Lovász extensions $f_r$ are convex and nonnegative. But the objective is in general nondifferentiable. To address this issue, we consider the dual of the QDSFM problem. A natural idea is to try to mimic the approach used for DSFM by invoking the characterization of the Lovász extension, $f_r(x) = \max_{y_r \in B_r}\langle y_r, x\rangle, \forall r$. However, this leads to a semidefinite programing problem for the dual variables $\{y_r\}_{r \in [R]}$, which is complex to solve for large problems. Instead, we establish a new dual formulation that overcomes this obstacle. The dual formulation hinges upon on the following key observation:

$$[f_r(x)]^2 = \max_{\phi_r \geq 0}\phi_r f_r(x) - \frac{\phi_r^2}{4} = \max_{\phi_r \geq 0}\max_{y_r \in \phi_r B_r}\langle y_r, x\rangle - \frac{\phi_r^2}{4}. \tag{5}$$

Let $y = (y_1, y_2, ..., y_R)$ and $\phi = (\phi_1, \phi_2, ..., \phi_R)$. Using equation (5), we arrive at

**Lemma 4.1.** The following optimization problem is dual to (3):

$$\min_{y,\phi} g(y,\phi) := \|\sum_{r\in[R]} y_r - 2Wa\|_{W^{-1}}^2 + \sum_{r\in[R]}\phi_r^2, \quad \text{s.t. } y \in \otimes_{r\in[R]}\phi_r B_r, \phi \in \otimes_{r\in[R]}\mathbb{R}_{\geq 0}. \tag{6}$$

By introducing $\Lambda = (\lambda_r) \in \otimes_{r\in[R]}\mathbb{R}^N$, the previous optimization problem can be rewritten as

$$\min_{y,\phi,\Lambda}\sum_{r\in[R]}\left[\|y_r - \frac{\lambda_r}{\sqrt{R}}\|_{W^{-1}}^2 + \phi_r^2\right], \text{ s.t. } y \in \otimes_{r\in[R]}\phi_r B_r, \phi \in \otimes_{r\in[R]}\mathbb{R}_{\geq 0}, \sum_{r\in[R]}\lambda_r = 2Wa. \tag{7}$$

The primal variables in both cases are recovered via $x = a - \frac{1}{2}W^{-1}\sum_{r\in[R]}y_r$.

Counterparts of the above results for the DSFM problem were discussed in Lemma 2 of [11]. However, there is a significant difference between [11] and the QDSFM problem, since in the latter setting we use a conic set constructed from base polytopes of submodular functions. More precisely, for each $r$, we define a convex cone $C_r = \{(y_r, \phi_r)|\phi_r \geq 0, y_r \in \phi_r B_r\}$ which gives the feasible set of the dual variables $(y_r, \phi_r)$. The optimization problem (7) essentially asks one to find the best approximation of an affine space in terms of a product cone $\otimes_{r\in[R]}C_r$, as opposed to a product polytope encountered in DSFM. Several algorithms have been developed for solving the DSFM problem, including the Douglas-Rachford splitting method (DR) [11], the alternative projection method (AP) [12] and the random coordinate descent method (RCD) [13]. Similarly, for QDSFM, we propose to solve the dual problem (6) using the RCD method exploiting the separable structure of the feasible set, and to solve (7) using the AP method. Although these similar methods for DSFM may be used for QDSFM, a novel scheme of analysis handling the conic structure is required, which takes all the effort in the rest of this section and the next section. Due to the page limitation, the analysis of the AP method is deferred to the full version of this paper. Also, it is worth mentioning that results of this work

can be easily extended for the DR method, as well as accelerated and parallel variants of the RCD method [13, 15].

**RCD Algorithm.** Define the projection $\Pi$ onto a convex cone $C_r$ as follows: for a given point $b$ in $\mathbb{R}^N$, let $\Pi_{C_r}(b) = \arg\min_{(y_r, \phi_r) \in C_r} \|y_r - b\|_{W^{-1}}^2 + \phi_r^2$. For each coordinate $r$, optimizing over the dual variables $(y_r, \phi_r)$ is equivalent to computing a projection onto the cone $C_r$. This gives rise to the RCD method summarized in Algorithm 1.

---

**Algorithm 1: RCD Solver for** (6)

---
0: For all $r$, initialize $y_r^{(0)} \leftarrow 0$, $\phi_r^{(0)}$ and $k \leftarrow 0$
1: In iteration $k$:
2:    Uniformly at random pick an $r \in [R]$.
3:    $(y_r^{(k+1)}, \phi_r^{(k+1)}) \leftarrow \Pi_{C_r}(2Wa - \sum_{r' \neq r} y_{r'})$
4:    Set $y_{r'}^{(k+1)} \leftarrow y_{r'}^{(k)}$ for $r' \neq r$

---

In Section 5, we describe efficient methods to compute the projections. But throughout the remainder of this section, we treat the projections as provided by an oracle. Note that each iteration of the RCD method only requires the computation of one projection onto a single cone. In contrast, methods such as DR, AP and the primal-dual hybrid gradient descent (PDHG) proposed in [29] used for SSL on hypergraphs [17], require performing a complete gradient descent and computing a total of $R$ projections at each iteration. Thus, from the perspective of iteration cost, RCD is significantly more efficient, especially when $R$ is large and computing $\Pi(\cdot)$ is costly.

The objective $g(y, \phi)$ described in (6) is not strongly convex in general. Inspired by the work for DSFM [13], in what follows, we show that this objective indeed satisfies a weak strong convexity condition, which guarantees linear convergence of the RCD algorithm. Note that due to the additional term $\phi$ that characterizes the conic structures, extra analytic effort is required than that for the DSFM case. We start by providing a general result that characterizes relevant geometric properties of the cone $\otimes_{r \in [R]} C_r$.

**Lemma 4.2.** Consider a feasible solution $(y, \phi) \in \otimes_{r \in [R]} C_r$ and a nonnegative vector $\phi' = (\phi'_r) \in \otimes_{r \in [R]} \mathbb{R}_{\geq 0}$. Let $s$ be an arbitrary point in the base polytope of $\sum_{r \in [R]} \phi'_r F_r$, and let $W^{(1)}, W^{(2)}$ be two positive diagonal matrices. Then, there exists a $y' \in \otimes_{r \in [R]} \phi'_r B_r$ such that $\sum_{r \in [R]} y'_r = s$ and

$$\|y - y'\|_{I(W^{(1)})}^2 + \|\phi - \phi'\|^2 \leq \mu(W^{(1)}, W^{(2)}) \left[ \|\sum_{r \in [R]} y_r - s\|_{W^{(2)}}^2 + \|\phi - \phi'\|^2 \right],$$

where

$$\mu(W^{(1)}, W^{(2)}) = \max \left\{ \sum_{i \in [N]} W_{ii}^{(1)} \sum_{j \in [N]} 1/W_{jj}^{(2)}, \frac{9}{4}\rho^2 \sum_{i \in [N]} W_{ii}^{(1)} + 1 \right\}. \tag{8}$$

As a corollary of Lemma 4.2, the next result establishes the weak strong convexity of $g(y, \phi)$. To proceed, we introduce some additional notation. Denote the set of solutions of problem (6) by

$$\Xi = \{(y, \phi) | \sum_{r \in [R]} y_r = 2W(a - x^*), \phi_r = \inf_{y_r \in \theta B_r} \theta, \forall r\}.$$

Note that this representation arises from the relationship between the optimal primal and dual solution as stated in Lemma 4.1. We denote the optimal value of the objective over $(y, \phi) \in \Xi$ by $g^* = g(y, \phi)$, and define a distance function $d((y, \phi), \Xi) = \sqrt{\min_{(y', \phi') \in \Xi} \|y - y'\|_{I(W^{-1})}^2 + \|\phi - \phi'\|^2}$.

**Lemma 4.3.** Suppose that $(y, \phi) \in \otimes_{r \in [R]} C_r$ and that $(y^*, \phi^*) \in \Xi$ minimizes $\|y - y^*\|_{I(W^{-1})}^2 + \|\phi - \phi^*\|^2$. Then

$$\|\sum_{r \in [R]} (y_r - y_r^*)\|_{W^{-1}}^2 + \|\phi - \phi^*\|^2 \geq \frac{d^2((y, \phi), \Xi)}{\mu(W^{-1}, W^{-1})}.$$

Based on Lemma 4.3, we can establish the linear convergence rate of the RCD algorithm.

**Theorem 4.4.** After $k$ iterations of Algorithm 1, we obtain a pair $(y^{(k)}, \phi^{(k)})$ that satisfies

$$\mathbb{E}\left[g(y^{(k)}, \phi^{(k)}) - g^* + d^2((y^{(k)}, \phi^{(k)}), \Xi)\right]$$
$$\leq \left[1 - \frac{2}{R[1 + \mu(W^{-1}, W^{-1})]}\right]^k \left[g(y^{(0)}, \phi^{(0)}) - g^* + d^2((y^{(0)}, r^{(0)}), \Xi)\right].$$

Theorem 4.4 implies that $O(R\mu(W^{-1}, W^{-1}) \log \frac{1}{\epsilon})$ iterations are required to obtain an $\epsilon$-optimal solution. Below we give the explicit characterization of the complexity for the SSL and PR problems with normalized Laplacian regularization as discussed in Section 3.

**Corollary 4.5.** Suppose that $W = \beta D$, where $\beta$ is a hyper-parameter, and $D$ is a diagonal degree matrix such that $D_{ii} = \sum_{r:\in[R], i\in S_r} \max_{S\subseteq V}[F_r(S)]^2$. Algorithm 1 requires an expected number of $O(N^2 R \max\{1, 9\beta^{-1}\} \max_{i,j\in[N]} \frac{D_{ii}}{D_{jj}} \log \frac{1}{\epsilon})$ iterations to return an $\epsilon$-optimal solution.

The term $N^2 R$ also appears in the expression for the complexity of the RCD method for solving the DSFM problem [14]. The term $\max\{1, 9\beta^{-1}\}$ implies that whenever $\beta$ is small, the convergence rate is slow. This makes sense: for example, in the PR problem (4), a small $\beta$ corresponds to a large $\alpha$, which typically implies longer mixing times of the underlying Markov process. The term $\max_{i,j\in[N]} \frac{D_{ii}}{D_{jj}}$ arises due to the degree-based normalization.

# 5 Computing the Projections $\Pi_{C_r}(\cdot)$

In this section, we provide efficient routines for computing the projection onto the conic set $\Pi_{C_r}(\cdot)$. As the procedure works for all values of $r \in [R]$, we drop the subscript $r$ for simplicity of notation. First, recall that

$$\Pi_C(a) = \arg\min_{(y,\phi)} h(y, \phi) \triangleq \|y - a\|_{\tilde{W}}^2 + \phi^2 \quad \text{s.t. } y \in \phi B, \phi \geq 0, \qquad (9)$$

where $\tilde{W} = W^{-1}$, and where $B$ denotes the base polytope of the submodular function $F$. Let $h^*$ and $(y^*, \phi^*)$ be the optimal value of the objective function and the argument that optimizes it, respectively. When performing projections, one only needs to consider the variables incident to $F$, and set all other variables to zero. For ease of exposition, we assume that all variables in $[N]$ are incident to $F$.

Unlike QDSFM, the DSFM involves the computation of projections onto the base polytopes of submodular functions. Two algorithms, the Frank-Wolfe (FW) method [30] and the Fujishige-Wolfe minimum norm algorithm (MNP) [31], are used for this purpose. Both methods assume cheap linear minimization oracles on polytopes and attain a $1/k$-convergence rate. The MNP algorithm is more sophisticated and empirically more efficient. Nonetheless, neither of these methods can be applied directly to cones. To this end, we modify these two methods by adjusting them to the conic structure in (9) and show that a $1/k-$convergence rate still holds. We refer to the procedures as *the conic MNP method* and *the conic FW method*, respectively. Here we focus mainly on the conic MNP method described in Algorithm 2, as it is more sophisticated. A detailed discussion of the conic FW method and its convergence guarantees can be found in the full version of this work.

The conic MNP algorithm keeps track of an *active set* $S = \{q_1, q_2, ...\}$ and searches for the best solution in its conic hull. Let us denote the cone of an active set $S$ as $\text{cone}(S) = \{\sum_{q_i\in S} \alpha_i q_i | \alpha_i \geq 0\}$ and its linear set as $\text{lin}(S) = \{\sum_{q_i\in S} \alpha_i q_i | \alpha_i \in \mathbb{R}\}$. Similar to the original MNP algorithm, Algorithm 2 also contains two level-loops: MAJOR and MINOR. In the MAJOR loop, we greedily add a new active point $q^{(k)}$ to the set $S$ obtained from the linear minimization oracle w.r.t. the base polytope (Step 2), and by the end of the MAJOR loop, we obtain a $y^{(k+1)}$ that minimizes $h(y, \phi)$ over $\text{cone}(S)$ (Step 3-8). The MINOR loop is activated when $\text{lin}(S)$ contains some point $z$ that guarantees a smaller value of the objective function than that of the optimal point in $\text{cone}(S)$, provided that some active points from $S$ may be removed. Compared to the original MNP method, Steps 2 and 5 as well as the termination Step 3 are specialized for the conic structure.

The following convergence result implies that the conic MNP algorithm also has a convergence rate of order $1/k$; the proof is essentially independent on the submodularity assumption and represents a careful modification of the arguments in [32] for conic structures.

| **Algorithm 2:** **The Conic MNP Method for Solving** (9) |
| --- |

**Input**: $\tilde{W}$, $a$, $B$ and a small positive constant $\delta$. **Maintain** $\phi^{(k)} = \sum_{q_i \in S^{(k)}} \lambda_i^{(k)}$

Choose an arbitrary $q_1 \in B$. Set $S^{(0)} \leftarrow \{q_1\}$, $\lambda_1^{(0)} \leftarrow \frac{\langle a, q_1 \rangle_{\tilde{W}}}{1 + \|q_1\|_{\tilde{W}}^2}$, $y^{(0)} \leftarrow \lambda_1 q_1$, $k \leftarrow 0$

1. Iteratively execute (**MAJOR LOOP**):
2.    $q^{(k)} \leftarrow \arg\min_{q \in B} \langle \nabla_y h(y^{(k)}, \phi^{(k)}), q \rangle_{\tilde{W}}$
3.    **If** $\langle y^{(k)} - a, q^{(k)} \rangle_{\tilde{W}} + \phi^{(k)} \geq -\delta$, then **break**; **Else** $S^{(k)} \leftarrow S^{(k)} \cup \{q^{(k)}\}$.
4.       Iteratively execute (**MINOR LOOP**):
5.          $\alpha \leftarrow \arg\min_{\alpha} \| \sum_{q_i^{(k)} \in S^{(k)}} \alpha_i q_i^{(k)} - a \|_{\tilde{W}}^2 + (\sum_{q_i^{(k)} \in S} \alpha_i)^2$, $z^{(k)} \leftarrow \sum_{q_i^{(k)} \in S} \alpha_i q_i^{(k)}$
6.          **If** $\alpha_i \geq 0$ for all $i$ then **break**
7.          **Else** $\theta = \min_{i:\alpha_i < 0} \lambda_i / (\lambda_i - \alpha_i)$, $\lambda_i^{(k+1)} \leftarrow \theta\alpha_i + (1-\theta)\lambda_i^{(k)}$,
8.            $y^{(k+1)} \leftarrow \theta z^{(k)} + (1-\theta)y^{(k)}$, $S^{(k+1)} \leftarrow \{i : \lambda^{(k+1)} > 0\}$, $k \leftarrow k+1$
9.    $y^{(k+1)} \leftarrow z^{(k)}$, $\lambda^{(k+1)} \leftarrow \alpha$, $S^{(k+1)} \leftarrow \{i : \lambda^{(k+1)} > 0\}$, $k \leftarrow k+1$

**Theorem 5.1.** Let $B$ be an arbitrary polytope in $\mathbb{R}^N$ and let $C = \{(y, \phi) | y \in \phi B, \phi \geq 0\}$ be the cone induced by the polytope. For some positive diagonal matrix $\tilde{W}$, define $Q = \max_{q \in B} \|q\|_{\tilde{W}}$. Algorithm 2 yields a sequence of $(y^{(k)}, \phi^{(k)})_{k=1,2,...}$ such that $h(y^{(k)}, \phi^{(k)})$ decreases monotonically. Algorithm 2 terminates when $k = O(N\|a\|_{\tilde{W}} \max\{Q^2, 1\}/\delta)$, with $h(y^{(k)}, \phi^{(k)}) \leq h^* + \delta\|a\|_{\tilde{W}}$.

Both the (conic) FW and MNP are approximate algorithms for computing the projections for generic polytopes $B$ and their induced cones. We also devised an algorithm of complexity $O(N \log N)$ that *exactly* computes the projection for polytopes $B$ arising in learning on (un)directed hypergraphs. A detailed description of the algorithm for exact projections is described in the full version of this paper.

# 6 Extension to mix-DSFM

With the tools to solve both QDSFM and DSFM problems, it is simple to derive an efficient solver for the following mix-DSFM problem: Suppose $\{F_r\}_{r \in [R_1 + R_2]}$ are a collection of submodular functions where $F_r \geq 0$ for $r \in [R_1]$. Let $f_r$ be the corresponding Lovász extension of $F_r$, $r \in [R_1 + R_2]$. We are to solve

$$\text{mix-DSFM:} \qquad \min_{x \in \mathbb{R}^N} \|x - a\|_W^2 + \sum_{r=1}^{R_1} [f_r(x)]^2 + \sum_{r=R_1+1}^{R_1+R_2} f_r(x) \qquad (10)$$

By using the same trick in (5) for the quadratic term, one may show the dual problem of mix-DSFM is essentially to find the best approximation of an affine space in terms of a mixture product of cones and base polytopes. Furthermore, all other related results, including the weak-strong duality of the dual, the linear convergence of RCD/AP and the $1/k$-rate convergence of the MNP/FW methods can be generalized to the mix-DSFM case via the same technique developed in this work.

# 7 Experiments

Our dataset experiments focus on SSL learning for hypergraphs on both real and synthetic datasets. For the particular problem at hand, the QDSFM problem can be formulated as follows

$$\min_{x \in \mathbb{R}^N} \beta\|x - a\|^2 + \sum_{r \in [R]} \max_{i,j \in S_r} \left(\frac{x_i}{\sqrt{W_{ii}}} - \frac{x_j}{\sqrt{W_{jj}}}\right)^2, \qquad (11)$$

where $a_i \in \{-1, 0, 1\}$ indicates if the corresponding variable $i$ has a negative, missing, or positive label, respectively, $\beta$ is a parameter that balances out the influence of observations and the regularization term, $\{W_{ii}\}_{i \in [N]}$ defines a positive measure over the variables and may be chosen to be the degree matrix $D$ with $D_{ii} = |\{r \in [R] : i \in S_r\}|$. Each part in the decomposition corresponds to one hyperedge. We compare eight different solvers falling into three categories: (a) our proposed general QDSFM solvers, *QRCD-SPE, QRCD-MNP, QRCD-FW and QAP-SPE*; (b) alternative solvers for the specific problem (11), including *PDHG* [17] and *SGD* [18]; (c) SSL solvers that do not use QDSFM as the objective, including *DRCD* [13] and *InvLap* [33]. The first three methods all

have outer-loops that execute RCD, but with different inner-loop projections computed via the *exact projection algorithm* for undirected hyperedges, or the generic MNP and FW. The QAP-SPE method uses AP in the outer-loop and exact inner-loop projections. PDHG and SGD are the only known solvers for the specific objective (11). DRCD is a state-of-the-art solver for DSFM and also uses a combination of outer-loop RCD and inner-loop projections. InvLap first transforms hyperedges into cliques and then solves a Laplacian-based linear problem. All the aforementioned methods, except InvLap, are implemented via C++ in a nonparallel fashion. InvLap is executed via matrix inversion operations in Matlab which may be parallelized. The CPU times of all methods are recorded on a 3.2GHz Intel Core i5. The results are summarized for 100 independent tests. When reporting the results, we use the primal gap ("gap") to characterize the convergence properties of different solvers. Additional descriptions of the settings and experimental results for the QRCD-MNP and QRCD-FW methods for general submodular functions can be found in the full version of this paper.

**Synthetic data.** We generated a hypergraph with $N = 1000$ vertices that belong to two equal-sized clusters. We uniformly at random generated 500 hyperedges within each cluster and 1000 hyperedges across the two clusters. Note that in higher-order clustering, we do not need to have many hyperedges within each cluster to obtain good clustering results. Each hyperedge includes 20 vertices. We also uniformly at random picked $l = 1, 2, 3, 4$ vertices from each cluster to represent labeled datapoints. With the vector $x$ obtained by solving (11), we classified the variables based on the Cheeger cut rule [17]: suppose that $\frac{x_{i_1}}{\sqrt{W_{i_1 i_1}}} \geq \frac{x_{i_2}}{\sqrt{W_{i_2 i_2}}} \geq \cdots \geq \frac{x_{i_N}}{\sqrt{W_{i_N i_N}}}$, and define $\mathcal{S}_j = \{i_1, i_2, ..., i_j\}$. We partition $[N]$ into two sets $(\mathcal{S}_{j^*}, \bar{\mathcal{S}}_{j^*})$, where

$$j^* = \arg\min_{j \in [N]} c(\mathcal{S}_j) \triangleq \frac{|S_r \cap \mathcal{S}_j \neq \emptyset, S_r \cap \bar{\mathcal{S}}_j \neq \emptyset\}|}{\max\{\sum_{r \in [R]} |S_r \cap \mathcal{S}_j|, \sum_{r \in [R]} |S_r \cap \bar{\mathcal{S}}_j|\}}.$$

The classification error is defined as (# of incorrectly classified vertices)$/N$. In the experiment, we used $W_{ii} = D_{ii}, \forall i$, and tuned $\beta$ to be nearly optimal for different objectives with respect to the classification error rates.

The top-left figure in Figure 1 shows that QRCD-SPE converges much faster than all other methods when solving the problem (11) according to the gap metric (we only plotted the curve for $l = 3$ as all other values of $l$ produce similar patterns). To avoid clutter, we postpone the results for QRCD-MNP and QRCD-FW to the full version of this paper, as these methods are typically 100 to 1000 times slower than QRCD-SPE. In the table that follows, we describe the performance of different methods with similar CPU-times. Note that when QRCD-SPE converges (with primal gap $10^{-9}$ achieved after 0.83s), the obtained $x$ leads to a much smaller classification error than other methods. QAP-SPE, PDHG and SGD all have large classification errors as they do not converge within short CPU time-frames. QAP-SPE and PDHG perform only a small number of iterations, but each iteration computes the projections for all the hyperedges, which is more time-consuming. The poor performance of DRCD implies that the DFSM is not a good objective for SSL. InvLap achieves moderate classification errors, but still does not match the performance of QRCD-SPE. Note that InvLap uses Matlab, which is optimized for matrix operations, and is hence fairly efficient. However, for experiments on real datasets, where one encounters fewer but significantly larger hyperedges, InvLap does not offer as good a performance as the one for synthetic data. The column "Average $100c(\mathcal{S}_{j^*})$" also illustrates that the QDSFM objective is a good choice for finding approximate balanced cuts of hypergraphs.

We also evaluated the influence of parameter choices on the convergence of QRCD methods. According to Theorem 4.4, the required number of RCD iterations for achieving an $\epsilon$-optimal solution for (11) is roughly $O(N^2 R \max(1, 9/(2\beta)) \max_{i,j \in [N]} W_{ii}/W_{jj} \log 1/\epsilon)$ (see the full version of this paper). We mainly focus on testing the dependence on the parameters $\beta$ and $\max_{i,j \in [N]} W_{ii}/W_{jj}$, as the term $N^2 R$ is also included in the iteration complexity of DSFM and was shown to be necessary given certain submodular structures [15]. To test the effect of $\beta$, we fix $W_{ii} = 1$ for all $i$, and vary $\beta \in [10^{-3}, 10^3]$. To test the effect of $W$, we fix $\beta = 1$ and randomly choose half of the vertices and set their $W_{ii}$ values to lie in $\{1, 0.1, 0.01, 0.001\}$, and set the remaining ones to 1. The two top-right plots of Figure. 1 show our results. The logarithm of gap ratios is proportional to $\log \beta^{-1}$ for small $\beta$, and $\log \max_{i,j \in [N]} W_{ii}/W_{jj}$, which is not as sensitive as predicted by Theorem 4.4. Moreover, when $\beta$ is relatively large ($> 1$), the dependence on $\beta$ levels out.

**Real data.** We also evaluated the proposed algorithms on three UCI datasets: *Mushroom*, *Covertype45*, *Covertype67*, used as standard datasets for SSL on hypergraphs [33, 17, 18]. Each dataset

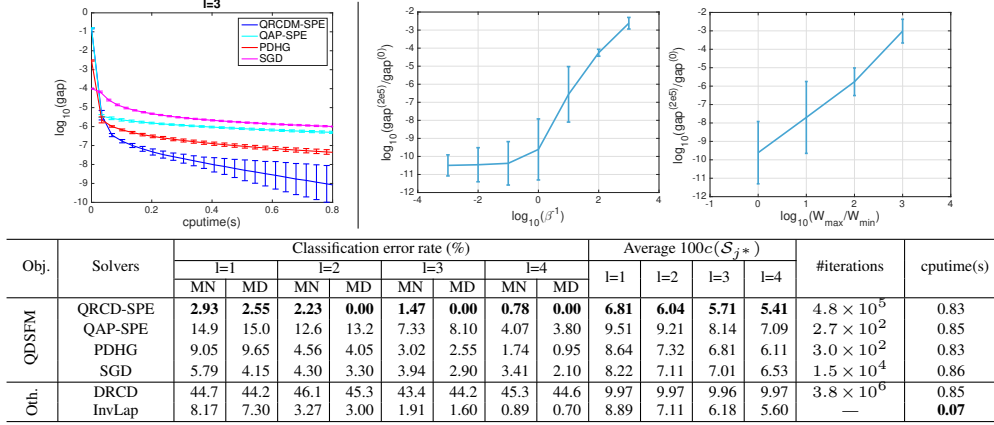

| Obj. | Solvers | Classification error rate (%) | | | | | | | | Average $100c(\mathcal{S}_{j*})$ | | | | #iterations | cputime(s) |
| | | l=1 | | l=2 | | l=3 | | l=4 | | l=1 | l=2 | l=3 | l=4 | | |
| | | MN | MD | MN | MD | MN | MD | MN | MD | | | | | | |
| QDSFM | QRCD-SPE | **2.93** | **2.55** | **2.23** | **0.00** | **1.47** | **0.00** | **0.78** | **0.00** | 6.81 | 6.04 | 5.71 | 5.41 | $4.8 \times 10^5$ | 0.83 |
| | QAP-SPE | 14.9 | 15.0 | 12.6 | 13.2 | 7.33 | 8.10 | 4.07 | 3.80 | 9.51 | 9.21 | 8.14 | 7.09 | $2.7 \times 10^2$ | 0.85 |
| | PDHG | 9.05 | 9.65 | 4.56 | 4.05 | 3.02 | 2.55 | 1.74 | 0.95 | 8.64 | 7.32 | 6.81 | 6.11 | $3.0 \times 10^2$ | 0.83 |
| | SGD | 5.79 | 4.15 | 4.30 | 3.30 | 3.94 | 2.90 | 3.41 | 2.10 | 8.22 | 7.11 | 7.01 | 6.53 | $1.5 \times 10^4$ | 0.86 |
| Oth. | DRCD | 44.7 | 44.2 | 46.1 | 45.3 | 43.4 | 44.2 | 45.3 | 44.6 | 9.97 | 9.97 | 9.96 | 9.97 | $3.8 \times 10^6$ | 0.85 |
| | InvLap | 8.17 | 7.30 | 3.27 | 3.00 | 1.91 | 1.60 | 0.89 | 0.70 | 8.89 | 7.11 | 6.18 | 5.60 | — | **0.07** |

Figure 1: Experimental results on synthetic datasets. Top-left: gap vs CPU-time of different QDSFM solvers (with an average $\pm$ standard deviation). Bottom: classification error rates & Average $100\ c(\mathcal{S}_{j*})$ for different solvers (MN: mean, MD: median). Top-right: the rate of a primal gap of QRCD after $2 \times 10^5$ iterations with respect to different choices of the parameters $\beta$ & $\max_{i,j\in[N]} W_{ii}/W_{jj}$.

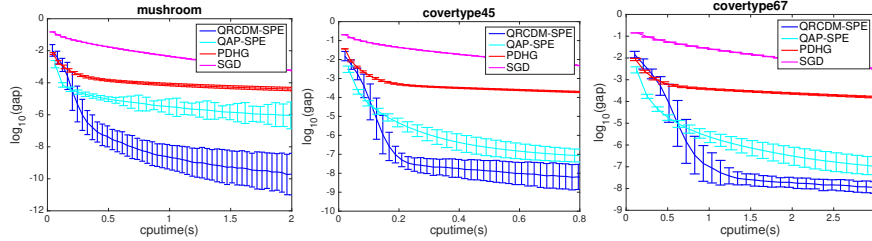

Figure 2: Convergence of different solvers for QDFSM over three different real datasets.

corresponds to a hypergraph model as described in [17]: entries correspond to vertices while each categorical feature is modeled as one hyperedge; numerical features are first quantized into 10 bins of equal size, and then mapped to hyperedges. Compared to synthetic data, in this datasets, the size of most hyperedges is much larger ($\geq 1000$) while the number of hyperedges is small ($\approx 100$). Previous works have been shown that fewer classification errors can be achieved by using QDSFM as an objective instead of DSFM or InvLap [17]. In our experiment, we focused on comparing the convergence of different solvers for QDSFM. We set $\beta = 100$ and $W_{ii} = 1$, for all $i$, and set the number of observed labels to 100, which is a proper setting as described in [17]. Figure. 2 shows the results. Again, the proposed QRCD-SPE and QAP-SPE methods both converge faster than PDHG and SGD, while QRCD-SPE performs the best. Note that we did not plot the results for QRCD-MNP and QRCD-FW as the methods converge extremely slowly due to the large sizes of the hyperedges. InvLap requires 22, 114 and 1802 seconds to run on the Mushroom, Covertype45 and Covertype67 datasets, respectively. Hence, the methods do not scale well.

# 8 Acknowledgement

The authors gratefully acknowledge many useful suggestions by the reviewers. This work was supported in part by the NIH grant 1u01 CA198943A and the NSF grant CCF 15-27636.

## Footnotes

[1]The code for QDSFM is available at https://github.com/lipan00123/QDSDM.

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
