[Reviews · NeurIPS 2018]

Reviewer 1



In this paper the authors consider the problem of minimizing sums of squares of Lovasz extensions. This is a convex problem and the authors derive its dual and propose several techniques for solving it that have a linear convergence rate. The consider problems over graphs - transductive learning and pagerank - and I am not aware of other problem classes that can be cast into this framework. What I find slightly confusing is why the authors posed the problem in terms of submodularity and Lovasz extensions, when, unless I'm mistaken, their theory and algorithms work for any polyhedral support function, not necessarily over base polyhedra. If I'm mistaken, it would be nice to point out what is the barrier to applying their methods beyond submodular polytopes. Moreover, unlike for submodular functions, there are no results about the quality of the rounded solution if we treat the problem as a continuous relaxation of the discrete problem. In the dual the authors obtain a dual containing cones derived from the base polytope and they develop algorithms for projecting onto them. They use Frank-Wolfe and a variant of it (Fujishige's min-norm algorithm is essentially a variant of fully corrective Frank-Wolfe). What is interesting is that they use FW which is classically defined only for compact domains and apply it to cones. They keep the same 1/k rate, and it would be quite useful if their analysis can be extended to other families of cones "generated" by polytopes. I unfortunately cannot comment much about the quality of the obtained solutions in the experimental section as I am not very familiar with the recent literature on transductive learning. I am interested why they did not compare against label propagation, which is a widely used method. Questions and remarks: 1) The Lovasz extension is not necessarily positive. I guess you assume F to map to [0, oo) instead of R. 2) Perhaps you should point out in line 126 that you are using the conjugate representation of the quadratic. 3) What would happen if you introduce linear terms in the objective? How would your algorithms change? 4) Why do you present the coordinate descent method in section 4, when you obtain the best results using AP? 5) Why not also plot the value of the function you are minimizing in the experimental section, which is more important than the duality gap. 6) It would be interesting to see how does accuracy change as a function of cpu time. I am not sure if having gaps of the order of 1e-10 makes a big difference, a the problems are defined using noisy data. 7) If I'm looking at it right, in Figure 1 it is the case that gaps of 1e-8 and 1e-9 can have a quite different classification performance (3% vs 9% error). Is the top figure not representative of the gaps you obtain for the lower table, or is something else going on? 8) What is the submodular function corresponding to (10)? It is not in your original table due to the presence of sqrt(w_ii) and sqrt(w_jj) in the denominator. 9) How does the accuracy look like for the real data? You're only showing duality gaps, and not comparing the accuracy of the existing work. ---- The rebuttal did address the questions that I had. To conclude, while I can not evaluate the impact on the transductive learning problems, I think this paper has several interest results on submodular optimization and the Frank-Wolfe algorithm that could be interesting to parts of the NIPS community. Hence, I am leaning towards accepting this submission.

Reviewer 2



The paper proposes to solve a new convex optimization problem termed quadratic decomposable submodular function minimization (QDSFM) where they first derive a dual formulation for the same and solve it using Random Coordinate Descent (RCD) establishing linear convergence. The conic projection step in RCD is computed via a modified Frank-Wolfe and min-norm point methods. The authors claim that their work is the first in solving QDSFM in its most general form. While I agree with their assertion, a lot of significant portion of their work follows from the previous established results in references [11], [13] and [14]. For example, the linear convergence of RCD is a known result. The key contribution of this work appears to the observation in equation (5) on which the dual formulation hinges upon. The authors cite references [4], [16] and [17] and state that employing regularization with quadratic terms offers improved predictive performance compared to p=1. On reading these citations it is not clear whether it is true only for the specific Lovasz extension considered in equation (10) or for the more general objective in equation (3) assuming the general convex f_r(x). Few minor typos: 1. Definition of y(S) in equation 2 is missing 2. The definition of f(x) in line 69 should be f(x) = max_{y \in B} ---------------------------------------------------------------------- The authors feedback addresses my concern regarding the contributions of this work. I agree with the authors that the weak convexity result established in Lemma 4.2 is new and an essential ingredient for their convergence results. However, the authors feedback do not provide examples of other decomposable submodular functions where their set up can be used beyond those considered in equation 10 which I believe is also a concern raised by Reviewer 1. I also do not see any comments on the quality of the rounded solution if we treat the problem as a continuous relaxation of the discrete problem.

Reviewer 3



Summary: This work proposes a generic class of optimization problems called quadratic decomposable submodular function minimization(QDSFM). They are helpful in solving page ranking, transductive learning problems. A dual is derived, which is solved using coordinated descent algorithms such as Alternating Projection or Random Co-ordinate Descent and the inner loop involves projection onto a cone. They have shown convergence guarantees and support their claim with experiments on synthetic and real datasets. Positives: - Overall, the paper is very well structured - The QDSFM is a novel contribution to literature, in my view. The algorithms, however are similar to the ones used to solve DSFM problems. - I have checked the proofs and the proofs seem correct. - They support the claim with synthetic and real data. Weaknesses/Requires Clarification: - Typos in the proof of lemma 4.1. They forgot the replace s by a. Similary, in the rest of the paper, it would be nice if the variables are used consistently. - In my practical experience, even for DSFM, the convergence is guarenteed only if the inner loops are exact projections. I believe, FW and MNP might not be useful in this case. There can be problem instances the approximate projection in innerloop does allow convergence to optimal. Could you add some comment/results for the QDSFM case. - It would be really helpful if a spell checker is run because there are some typos in the paper. (eg: Muchroom in line 253) Update after Author response: I agree that FW is approximate and MNP is exact. I apologize to have mentioned MNP as approximate. However, what I meant was FW is less expensive but approximate while MNP is exact but its complexity depends very much on the function. Generic MNP oracles are expensive. However, it does make sense that we need exact oracles in the inner loop for convergence of the outerloop.